# Black Rock City versus Manhattan: An economist's view

**John Yinger** *

Syracuse University, Syracuse, New York, United States of America

* jyinger@maxwll.syr.edu

## Abstract

Urban street networks take many forms, from the circular streets in Black Rock City (which is built and removed every year as part of the Burning Man Festival) to the streets and avenues in the Manhattan grid. This paper compares the traits of cities with different street networks using the tools of urban economics. When both cities have commuting arteries of the same length, cities with circular streets have higher population densities unless access to these arteries is expensive and the number of arteries is large. Cities with arteries set at a 45° angle to the grid have smaller population densities under all circumstances.

**Data Availability Statement:** This is a conceptual paper. No data sets were used. Calculations are fully explained in the paper.

**Funding:** The author received no specific funding for this work.

## 1. Introduction

Every year tens of thousands of people gather in the Nevada desert for the countercultural Burning Man Festival. This festival may appear to be a far cry from urban economics, but in fact the organizers of this festival design and build an entire city, called Black Rock City. They draw a map of the city's street network and reproduce this map on the desert itself. They designate camping sites served by this street network and develop rules for allocating these sites to festival participants. This paper uses the design of Black Rock City, which has circular streets, as a starting point to explore, from an economist's point of view, the impact of street network design on a city's broad characteristics. An alternative city design, which also is considered in this paper, is the grid system made famous by the 1811 plan for the island of Manhattan in New York City. These two designs have a similar scale. The diameter of Burning Man City (2.51 miles in 2019) is about the same as the width of Manhattan at its widest point (2.3 miles).

This paper compares the area, population, and population density of cities with different street networks, including circles and grids with and without commuting arteries. This analysis is conducted through the lens of urban economics based on mathematical models of urban residential structure. City design raises many other issues, of course. Planners for Black Rock City, for example, argue that "[t]hese circles and resulting intersections were intentionally designed for social interaction, not for transportation or commerce. It's impossible to go from point A to point B without having several serendipitous moments with strangers" [1]. In contrast, the appointed commissioners for the 1811 New York City grid, argued that "[a] city is to be composed principally of the habitations of men and strait-sided and right-angled houses are the most cheap to build and the most convenient to live in" [2]. These issues of microdesign are not considered here. The analysis in this paper also has nothing to contribute to the

**Competing interests:** The author has no competing interests.

analysis of a "circular" city defined as one "that eliminates waste, keeps goods and their ingredients in use and regenerates natural systems" [3].

This paper begins (Section 2) by using the tools of urban economics to compare two simple cities with different street patterns. Three more complex cities, which have the same commuting arteries but different street networks, are compared in Section 3, which is the analytical core of the paper. Section 4 discusses some possible extensions of the analysis, and Section 5 summarizes the paper's main lessons.

## 2. Comparing simple urban models

A basic monocentric model of urban residential structure, or urban model for short, is built around distance to the central business district (CBD), $u$. (I do not consider models based on commuting time, because they cannot be drawn on a map.) This approach implicitly assumes that the transportation system consists of a large number of streets that take the form of rays into the CBD. The cost of reaching a ray is ignored. I call a city with these assumptions "Ray City." This approach, which is designed to approximate an actual transportation system, results in circular iso-commuting-cost lines (iso-cost lines for short) for households. Assuming that the urban area (henceforth called a "city" for conciseness) is a full circle, the amount of land along an iso-cost line, $I_{Ray}\{u\}$, equals $2\pi u$, where curly brackets enclose the arguments of a function. Let $\bar{u}$ indicate the outer edge of the city. Then the amount of land in the city, $L_{Ray}$, is $\pi\bar{u}^2$. The S1 Appendix includes a table of notation.)

For the purposes of this paper, the expression $2\pi$ is called the "land constant." With Cobb-Douglas utility and production functions, the total population in Ray City, $N_{Ray}$, equals this land constant multiplied by an expression, say $N^*\{\bar{u}\}$, that does not generally depend on the design of the street network; that is, $N_{Ray} = (2\pi)N^*\{\bar{u}\}$, where

$$N_{Ray} = \left(\frac{\bar{R}2\pi}{t}\right)\left(\frac{Y^{b+1}}{t(b+1)(Y-t\bar{u})^b} - \frac{Y-t\bar{u}}{t(b+1)} - \bar{u}\right) \equiv 2\pi N^*\{\bar{u}\} \tag{1}$$

In this well-known equation (see, e.g., the derivation in [4], $\bar{R}$ is the agricultural rental rate for land, $t$ is round-trip per mile commuting cost, $Y$ is household income, $b = 1/(a\alpha)$, $a$ is the exponent on land in the housing production function, and $\alpha$ is the coefficient on housing in the household utility function. Dollar values are expressed in per-day terms. Because average population density, $D$, equals population divided by area, $D_{Ray} = N_{Ray}/ L_{Ray} = 2N^*\{\bar{u}\}/\bar{u}^2$.

An alternative street network in an urban model, first proposed by [5], is a street grid. Travel along a street grid is termed "Manhattan distance" by [6]. Analysis of urban models with this form, which I call "Manhat City," can be found in [7] and [8]. With the same commuting cost along horizontal and vertical streets, an iso-cost line in this case is a tilted square. The length of the iso-cost line that touches the vertical street $u$ miles from the CBD is the circumference of the square that goes through that point. Hence, $I_{Manhat}\{u\} = 4\sqrt{u^2 + u^2} = 4\sqrt{2}u$, and the land constant is $4\sqrt{2}$. Total land in the city is the area of the iso-cost square at $\bar{u}$ or $L_{Manhat} = 2\bar{u}^2$. With the same assumptions about households, housing production, and transportation costs as with Ray City, $N_{Manhat} = 4\sqrt{2}\, N^*\{\bar{u}\}$ and $D_{Manhat} = N_{Manhat}/L_{Manhat} = 2\sqrt{2}N^*\{\bar{u}\}/\bar{u}^2$.

These results allow us to compare the two urban areas, holding the maximum distance from the center, $\bar{u}$, constant. First, Ray City is 57.1 percent larger in area: $L_{Ray}/L_{Manhat} = \pi\bar{u}^2/(2\bar{u}^2) = \pi/2 = 1.571$. Second, Ray City city contains 11.1 percent more people: $N_{Ray}/N_{Manhat} = 2\pi/(4\sqrt{2}) = 1.111$. Third, Ray City has 29.3 percent fewer people per square mile: $D_{Ray}/D_{Manhat} = (2N*/\bar{u}^2)/(2\sqrt{2}N^*/\bar{u}^2) = 1/\sqrt{2} = 0.7071$.

An alternative comparison is to hold population constant across the two cities. The assumed parameter values for these calculations and the method, which is based on the derivative of Eq (1) with respect to $\bar{u}$, are presented in the S1 Appendix. Comparisons across cities with different transportation networks are similar with other assumptions. These simulations begin with initial $\bar{u}$ equal to 30 miles in both cities, a population of 437,822 in Manhat City and of 486,298 (11.1 percent larger!) in Ray City. To make the populations the same, the value of $\bar{u}$ in Manhat City would have to increase to 30.7 miles, which is an increase of 2.3 percent. With this larger value for $\bar{u}$ in Manhat City, its area would increase by 4.7 percent. This change lowers but does not eliminate the size difference between the two cities; Ray City is still 50 percent larger, with a 33 percent lower density. Compared to an equal-population Manhat City, Ray City is more spread out and less dense.

The models in this paper all assume that the CBD is a point. At the cost of more algebraic complexity, the CBD could instead be given a fixed radius. With a common radius for all cities, this extension would have little or no impact on city comparisons. As a result, the simpler approach is maintained here. A more complicated approach is to model the production of an export good in the CBD, complete with firm demand for land. This approach, which can be found in [9] and [10], must solve for the CBD radius. This step is not taken in this paper.

## 3. Comparing urban models with arteries

### 3.1. Basic analysis

The results in the previous section are intriguing but not very satisfying, because they are based on a radial city model in which travel distance to radial streets or arteries is ignored and on a grid-city model with no commuting arteries. This section develops similar across-model comparisons with more realistic models. These models incorporate commuting arteries, which were introduced by [5] and which appear in [4, 9, 11, 12], and [13].

This paper compares three types of cities with commuting arteries. The first, which I call "Circle City," builds on the transportation network and associated urban model in the classic article by Anas and Moses [9]. This network consists of circular streets and commuting arteries, which could be either highways or commuter rail. This type of network is also considered by [12] and [13].

As shown in Fig 1, this network is essentially the design for Black Rock City, although in that case the "arteries" are not distinguished by relatively high commuting speeds [14]. (Indeed, the speed limit for cars in Black Rock City on streets and "arteries" is only 5 MPH [1].) Black Rock City does have a center, dominated by a large statue called The Man, where numerous galleries and activities attract the city's inhabitants. However, the "market" for land in Black Rock City is unusual. "In the early 2000s theme camps—elaborate enclaves built by groups of longtime Burners [i.e. Burning Man Festival participants]—filled the innermost of the inhabited circles, with the best views of the Man. A 2005 rezoning spread them out along the radial streets" [15]. Both of these allocation methods hint at the logic of an urban model, as groups that have invested the most are placed in the sites with the best access to the center. Nevertheless, the analysis in this paper is intended to shed light on cities with the street network in Fig 1, not on Black Rock City itself.

[9] uses this setup to study the impact of mode choice on household location. This step is accomplished by adding the equivalent of Ray City, that is, a second transportation mode using unspecified secondary streets that allow commuters to travel radially to the CBD, at least as a first approximation. [9] then determines the boundaries between the residential areas in which commuters use the arteries and those in which they use the secondary streets. To facilitate comparisons across cities with different transportation networks, this paper focuses on a

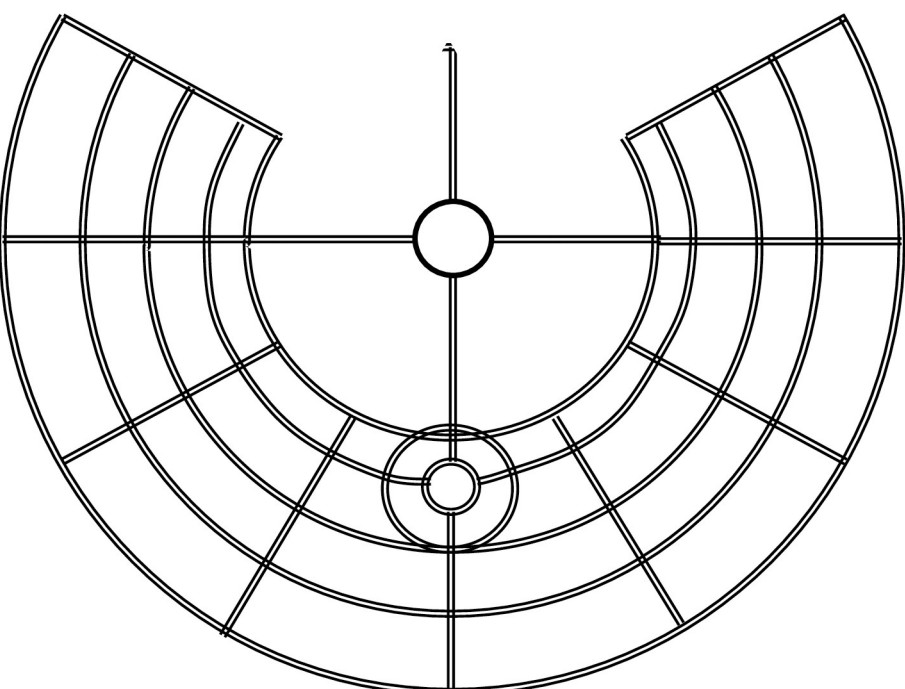

**Fig 1. Illustrative streets in Black Rock City, 2019.**

"Circle City" in which every commuter uses a radial artery. I start with a transportation network that has with four equally spaced radial arteries originating in the CBD. This is case b in Anas and Moses. The impact of additional arteries is considered in Section 3.3.

The iso-cost line for the Anas/Moses transportation network that goes through a point on an artery $u$ miles from CBD is defined by:

$$t_a(u - u') = t_s\rho\{u'\} = t_s u'\theta\{u\} \tag{2}$$

In this equation, $t_a$ is per-mile transportation costs along the artery, $t_s$ is the comparable cost along a circular street, $\rho$ is distance along a circular street, and $\theta$ is this distance expressed in radians. This equation identifies locations where the cost of traveling along a circular street to reach a commuting artery at location $u'$ equals the cost of traveling along the artery from $u$ to $u'$.

These concepts are illustrated in Fig 2, which depicts the positive quadrant of a city with this Anas/Moses transportation system. The axes in this figure correspond to two of the city's four arteries. Note that this city is characterized by a commuting-shed boundary; households living above this boundary commute to the vertical artery and households living below it commute to the horizontal artery. In addition, the circular streets are incomplete at many distances from the CBD, as illustrated by the narrowing of the illustrative streets in Fig 2, because, in this model and others, the residential area extends the farthest directly along the paths of the relatively high-speed arteries. In this figure, commuting costs on the artery are assumed to be half as large as commuting costs along the circular streets.

The first step in understanding Circle City is to derive the shape and length of an iso-cost line, that is, a set of locations with equal transportation costs. A segment of the iso-cost line is defined by the distance from point A to point B in Fig 2. The length of the iso-cost line that intersects the vertical artery $u$ miles from the CBD, $I_{Circle}\{u\}$, equals this integral evaluated

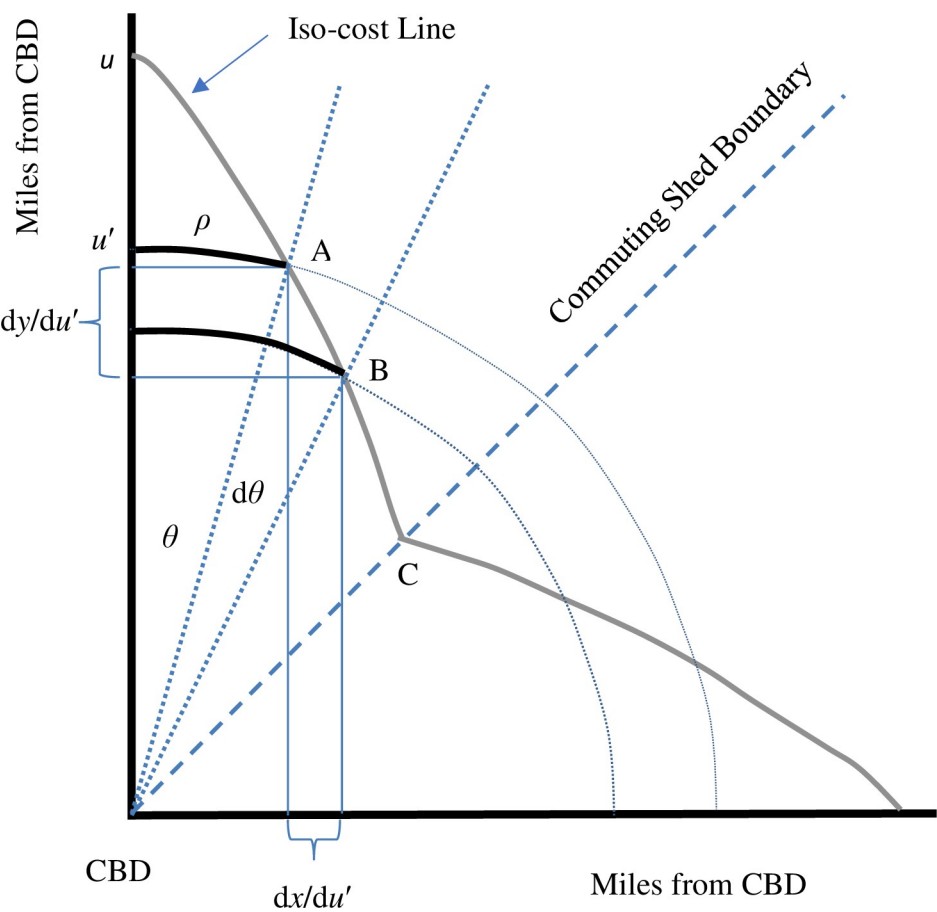

**Fig 2. Solving for the Anas-Moses iso-cost line.**

from the shed boundary to the y-axis. With four arteries, this integral must be multiplied by eight, because this formula applies to one of eight segments that make up the city. This integral is spelled out in the S1 Appendix.

Arteries also have been added to urban models with a street grid by [8, 11, 13], and [16]. This paper focuses on arteries that go to the CBD. Other arrangements are possible, of course. See the plan for Washington, D.C. [17] and models of arteries that do not go to the CBD in [5] and [8]. Consider first a transportation network with a street grid that is lined up with four commuting arteries to the CBD. See Fig 3. I call this type of city "Grid City." Assuming that the cost of travel is the same on both vertical and horizontal streets, the iso-cost line through $u$ for this transportation system in the positive quadrant is defined by:

$$y = \frac{t_a u - t_s x}{t_a} = u - \frac{t_s}{t_a} x = u - \frac{x}{\bar{t}} \tag{3}$$

where $\bar{t} \equiv t_a/t_s$ and $x$ and $y$ are coordinates. Hence an iso-cost line is a straight line with slope $-1/\bar{t}$. Moreover, the shed boundary is defined by $y = x$. These concepts are illustrated in Fig 3. The area of the city is defined by the triangle $[(0,0), (0,u), (x^*,y^*)]$ when $u = \bar{u}$. The length of the outer iso-cost line is the hypotenuse of this triangle. In both cases the result must multiplied by the number of segments (eight in this case). The S1 Appendix derives these results and proves that $I_{Grid}\{u\} = (\phi_{Grid})u$, where $\phi_{Grid}$ is a constant.

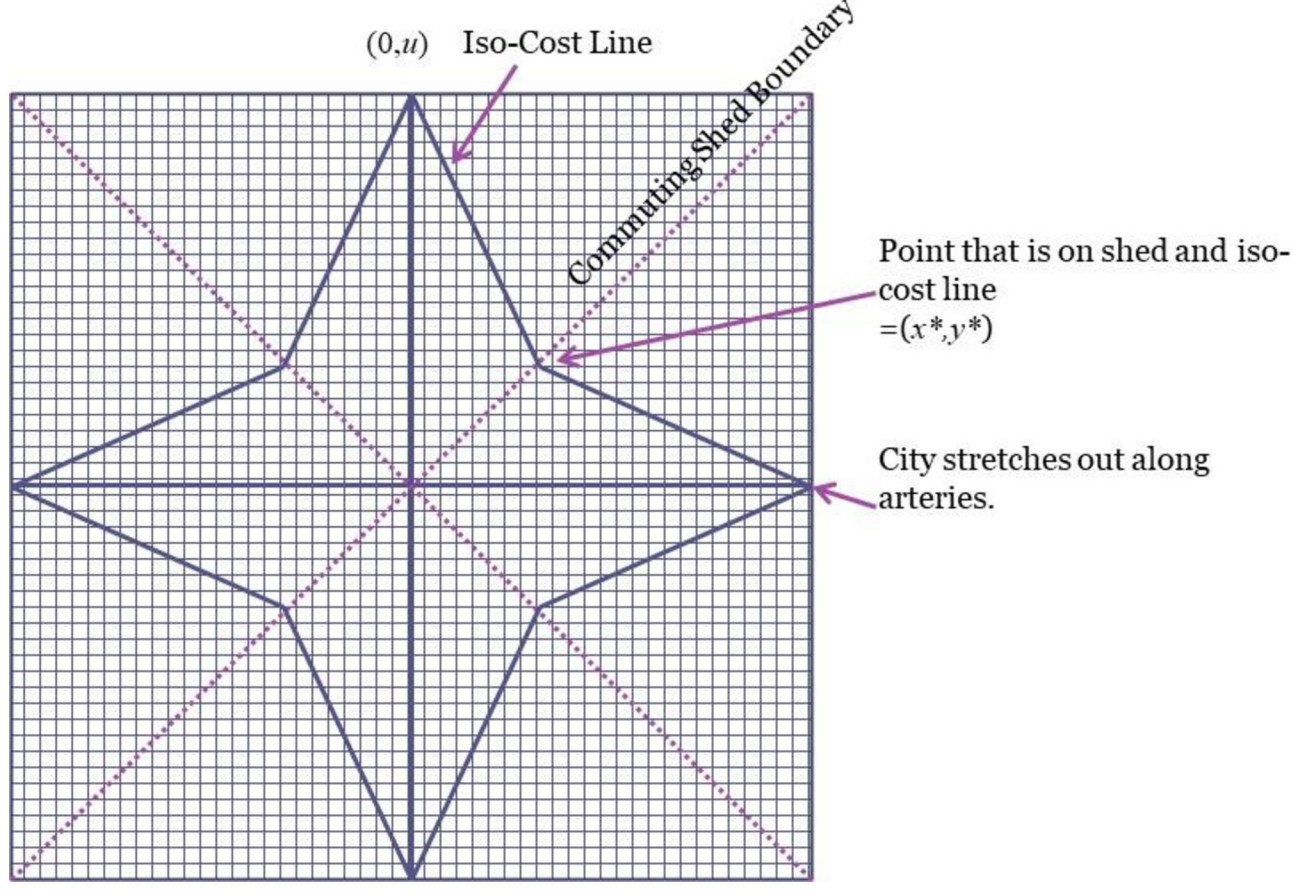

**Fig 3. Shed boundaries and iso-cost lines with a grid.**

Fig 4 compares the iso-cost lines for Circle City and Grid City. The iso-cost line segments in Circle City are curved, whereas the iso-cost line in Grid City is a series of straight-line segments. The formula for $I_{Circle}\{u\}$, which is derived in the S1 Appendix, is a complicated function of $u$, whereas $I_{Grid}\{u\} = (\phi_{Grid})u$, where $\phi_{Grid}$ is a constant. Numerical simulation reveals, however, that $I_{Circle}\{u\}$ closely approximates this form; that is, $I_{Circle}\{u\} \approx (\phi_{Circle})u$, With four arteries and $\bar{t} = 0.5$, for example, $(\phi_{Circle})$ exceeds $\phi_{Grid}$ by close to 8 percent for all $u$. As discussed below, this property of the iso-cost lines is useful for determining city populations.

Another possibility, also considered by Yinger [8], is that the commuting arteries are not lined up with the grid. The plan for Washington, D.C., which is largely based on the 1791 plan by Pierre Charles L'Enfant (see [17]), defines the U.S. Capitol as the city center and the Mall as one of the "arteries" emanating from this center, most of which are not aligned with the street grid. Many arteries in this plan go to other centers, of course, but only one center is considered here. This arrangement leads us to consider "Diag City," which has arteries at a forty-five-degree angle from the grid. See Fig 5, in which the diagonal arteries are treated as the (tilted) $x$ and $y$ axes. One segment of Diag City has the area of the triangle $[(0, 0), (0, u), (x^*, y^*)]$ evaluated at $u = \bar{u}$. Moreover, the length of an iso-cost segment is the hypotenuse of the triangle $[(0, y^*), (0, \bar{u}), (x^*, y^*)]$, The formulas for these quantities and for $I_{Diag}\{u\} = (\phi_{Diag})u$ are derived in the S1 Appendix.

One key feature of Diag City is that its shape could be a multi-pointed star, as in Fig 5, or a regular polygon. [8] shows that with four arteries an octagon shape arises if $\bar{t} \geq 1/\sqrt{2} = 0.7071$.

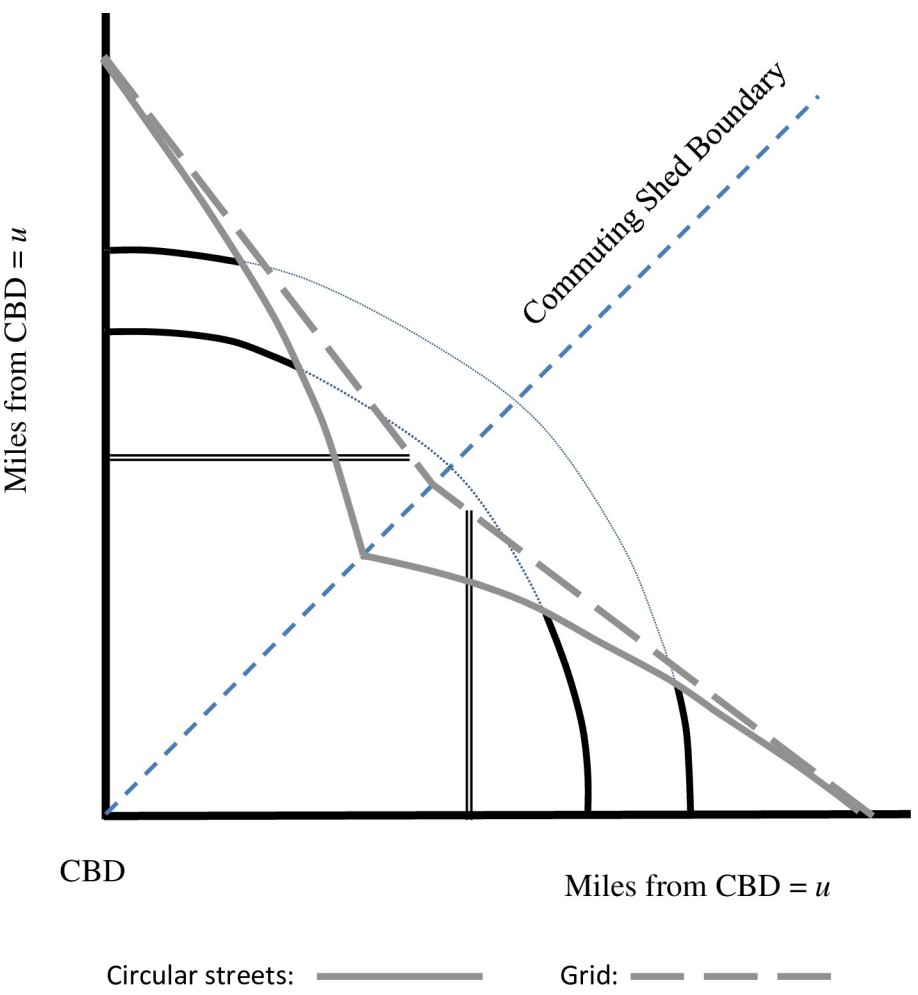

**Fig 4. Iso-cost lines with circular streets and a grid.**

The S1 Appendix derives a more general formula, which indicates a decline in the minimum $\bar{t}$ that leads to a polygon as the number of arteries goes up. With sixteen arteries, for example, the polygon shape arises whenever $\bar{t} \geq 0.235$. Thus, a polygon shape is more likely to arise in cities with more arteries or with relatively low-cost travel on streets, all else equal.

The objective of this paper is to compare the areas, populations, and average population densities of Circle City, Grid City, and Diag City. The areas are the spaces on a map enclosed by the outermost iso-cost lines, that is, the ones associated with $\bar{u}$. The relevant formulas are in the S1 Appendix. The calculations in this paper set $t_a = 1.5$ and alter $\bar{t}$ by altering the value of $t_s$. Changing $t_a$ obviously would alter the characteristics of a city, but it would have little or no impact on inter-city comparisons.

The population of a city can be determined using a theorem in Yinger [8], namely that Eq (1) determines city population for a city with any road network, so long as the land supply at location $u$ is proportional to $u$, that is, if $I\{u\} = \phi^* u$, where $\phi^*$ is a constant. As shown earlier, this formulation applies to the three cities in the paper. The application of this theorem to these cities must reflect the balance between two factors. First, lowering $\bar{t}$ usually increases the length of a city's iso-cost lines, which leads to an increase in population. The second factor, which I call the "squish" factor, is that lowering $\bar{t}$ reduces a city's area and thereby reduces its

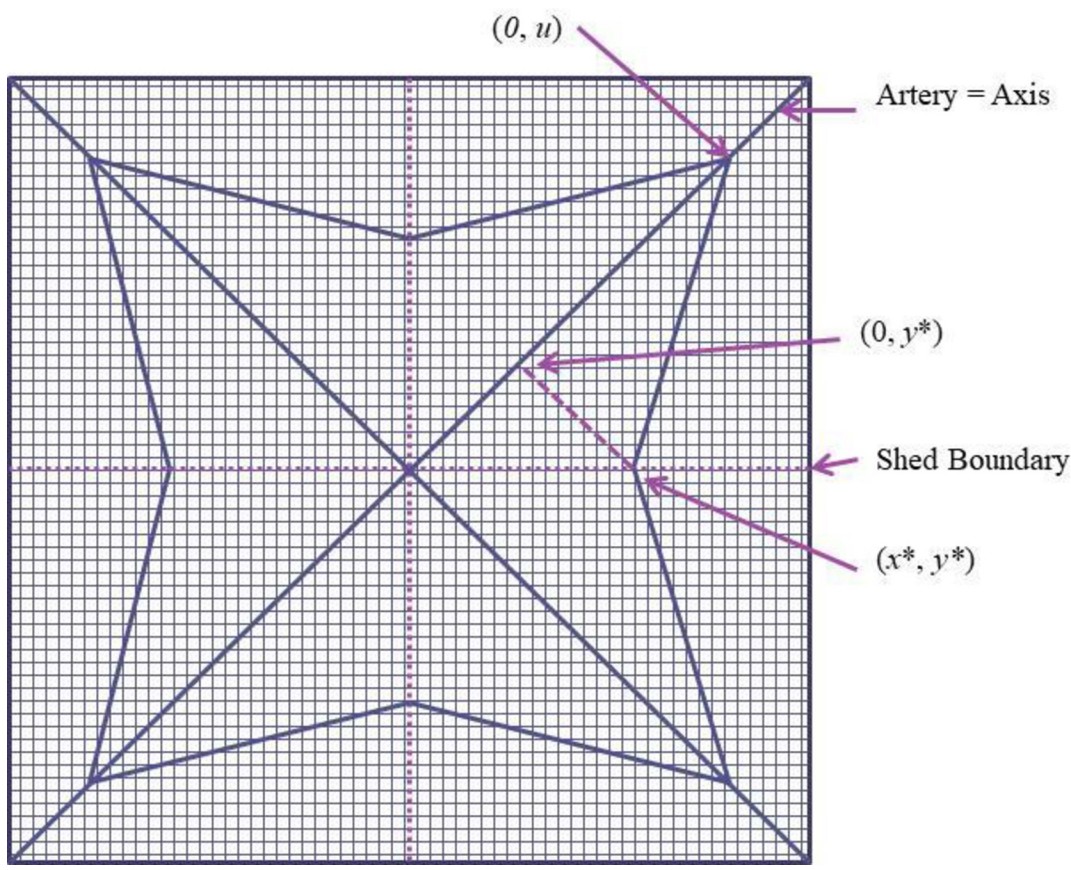

**Fig 5. Shed boundaries and iso-cost lines with diagonal arteries.**

population. Formulas to account for these two factors are described in the S1 Appendix. Average population density is, of course, the ratio of total population to total area.

## 3.2. Comparisons that account for relative commuting costs on different routes

The relative commuting costs along arteries and streets may vary considerably across cities. This section shows how changes in this relative speed affect urban form and how they affect a comparison of the three city types in this paper.

Consider first the case of area. The first four columns of Table 1 and the left-most points in Fig 6 describe cities with four arteries. As $\bar{t}$ increases, that is, as the cost of travel on streets moves closer to the cost of travel on the artery, the area of a city increases regardless of its transportation network design. An increase in $\bar{t}$ represents an increase in transportation speed on streets (equivalent here to a decrease in transportation costs), which encourages households to locate farther away from arteries and thereby to increase the physical size of the city. These four columns reveal that Grid City has the largest area among the three cities when $\bar{t}$ is small, Grid City and Diag City have the same area when $\bar{t} = 0.414$, and Diag City has the largest area at higher values of $\bar{t}$. Circle City does not have the largest area in any of these columns.

A different pattern emerges with population. With four arteries, Diag City has the largest population when travel on streets is relatively fast and the city takes the shape of an octagon, but it has the smallest population when street travel is relatively slow. Circle City and Grid City

**Table 1. A comparison of three city types.**

| | 4 Arteries | | | | 16 Arteries | | | |
|---|---|---|---|---|---|---|---|---|
| PARAMETERS | | | | | | | | |
| $t_a$ | 1.500 | 1.500 | 1.500 | 1.500 | 1.500 | 1.500 | 1.500 | 1.500 |
| $t_s$ | 6.000 | 3.621 | 3.000 | 1.579 | 6.000 | 3.621 | 3.000 | 1.579 |
| $\bar{t}$ | 0.250 | 0.414 | 0.500 | 0.950 | 0.250 | 0.414 | 0.500 | 0.950 |
| AREA (SQUARE MILES) | | | | | | | | |
| Circle City | 683 | 976 | 1100 | 1548 | 1584 | 1918 | 2030 | 2343 |
| Grid City | 720 | 1054 | 1200 | 1754 | 1595 | 1935 | 2049 | 2368 |
| Diag City | 636 | 1054 | 1273 | 2418 | 1487 | 1935 | 2100 | 2611 |
| POPULATION | | | | | | | | |
| Circle City | 196,814 | 280,322 | 315,315 | 441,864 | 361,867 | 453,664 | 487,489 | 594,548 |
| Grid City | 175,129 | 256,470 | 291,881 | 426,596 | 387,996 | 470,678 | 498,421 | 576,084 |
| Diag City | 118,474 | 196,294 | 236,948 | 450,200 | 276,818 | 360,241 | 391,022 | 485,998 |
| DENSITY (PEOPLE PER SQUARE MILE) | | | | | | | | |
| Circle City | 288 | 287 | 287 | 285 | 229 | 237 | 240 | 254 |
| Grid City | 243 | 243 | 243 | 243 | 243 | 243 | 243 | 243 |
| Diag City | 186 | 186 | 186 | 186 | 186 | 186 | 186 | 186 |
| CHANGE IN CITY OUTER EDGE ($\bar{u}$) NEEDED TO MATCH LARGEST POPULATION (MILES) | | | | | | | | |
| Circle City | - | - | - | 0.12 | 1.48 | 1.28 | 1.08 | - |
| Grid City | 0.14 | 0.19 | 0.21 | 0.34 | - | - | - | 0.27 |
| Diag City | 0.96 | 1.06 | 1.01 | - | 1.61 | 1.60 | 1.55 | 1.57 |
| ARTERY LENGTH ($\bar{u}$) NEEDED TO REACH A POPULATION OF 100,000 | | | | | | | | |
| Circle City | 24.9 | 22.2 | 21.4 | 19.1 | 20.4 | 18.9 | 18.5 | 17.2 |
| Grid City | 25.7 | 22.9 | 22.0 | 19.3 | 20.0 | 18.7 | 18.3 | 17.4 |
| Diag City | 28.7 | 24.9 | 23.5 | 19.0 | 22.4 | 20.5 | 19.9 | 18.5 |
| ARTERY LENGTH ($\bar{u}$) NEEDED TO REACH A POPULATION OF 500,000 | | | | | | | | |
| Circle City | 37.4 | 34.6 | 33.7 | 31.0 | 32.6 | 30.8 | 30.2 | 28.7 |
| Grid City | 38.3 | 35.3 | 34.3 | 30.9 | 32.0 | 30.5 | 30.1 | 28.9 |
| Diag City | 41.3 | 37.4 | 35.9 | 31.3 | 34.7 | 32.6 | 32.0 | 30.3 |

Source: Author's calculations based on the urban models and assumed parameter values in the S1 Appendix. The value of $\bar{t}$ is the ratio of $t_a$, commuting costs per mile along an artery, to $t_s$, commuting costs along a street. Comparisons across cities are similar with other parameter values.

have similar populations, but Circle City's population is higher regardless of the value of $\bar{t}$. These results also appear in Fig 7, where the four-artery case appears at the left edge of the graph.

These results lead to clear statements about average population density with four arteries. Circle City has the highest density, Grid City has density only slightly lower than that of Circle City, and Diag City has the lowest density. Moreover, the densities of the two cities with grids do not vary with the relative cost of commuting on arteries versus streets.

Finally, the changes in $\bar{u}$ needed to bring all cities up to the same population are not large. The largest change in the first four columns of Table 1 is the change required to bring the population of Diag City up to the population of Circle City when $\bar{t} = 0.414$. In this case the value of $\bar{u}$ in Diag City would have to increase by 1.22 miles or 1.22/30 = 4.1 percent.

## 3.3. Comparisons that account for the number of arteries

A second key dimension for analyzing these three cities is the number of arteries. Models with more than four arteries are discussed by [8, 9], and [13]. Actual cities are unlikely to have only

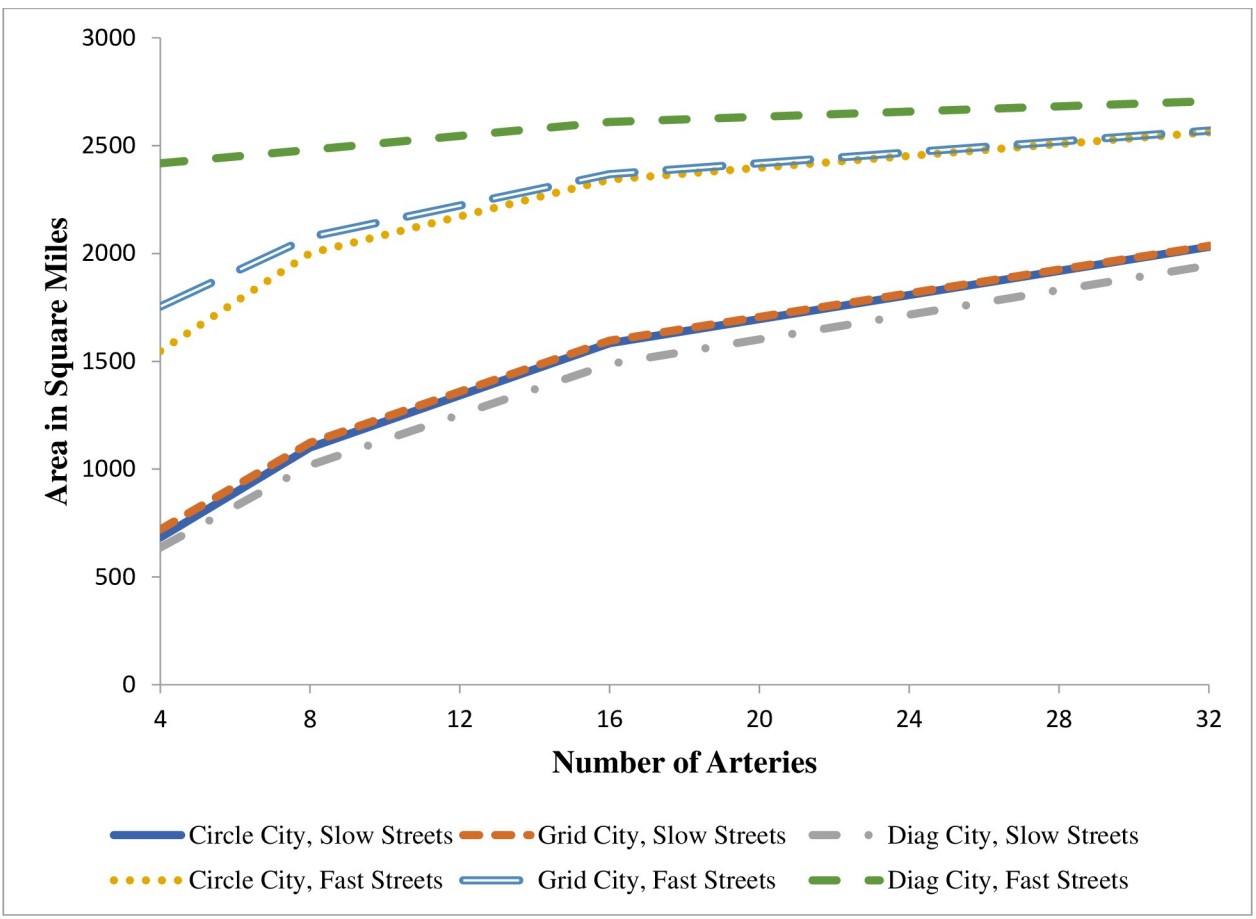

**Fig 6. City area and number of arteries.**

four arteries, of course. The portion of Washington, D.C. around the U.S. Capitol, for example, has ten arteries, including the Mall. The 2019 design for Black Rock City contained seventeen arteries in two-thirds of a circle [14].

Because of the symmetry provided by circular streets, the addition of arteries to Circle City is straightforward. In the case of sixteen arteries, for example, the city can be divided into thirty-two identical segments, and the maximum distance between arteries at the outer edge of the city is 11.8 miles, compared to 47.1 miles with four arteries. Because a city extends farther along high-speed arteries than along streets, adding arteries leads to "suburbanization," that is, to an outward shift in the distribution of a city's population by distance from the CBD. This shift takes the form of more locations that extend out to $\bar{u}$ along arteries and smaller indentations in the city boundary between arteries.

This link between arterial highways and an outward shift in population is not quite the same thing as the suburbanization studied by [13] and [18]. This analysis is based on closed urban models, which assume that highway changes arise in all cities so that a city's population does not change. This approach is appropriate for studying a national phenomenon. My analysis is based on open urban models, which assume that highway changes arise in only one city in a system, so that these changes may lead to inter-city migration and population change. This approach is appropriate when one city adds arteries but other cities do not.

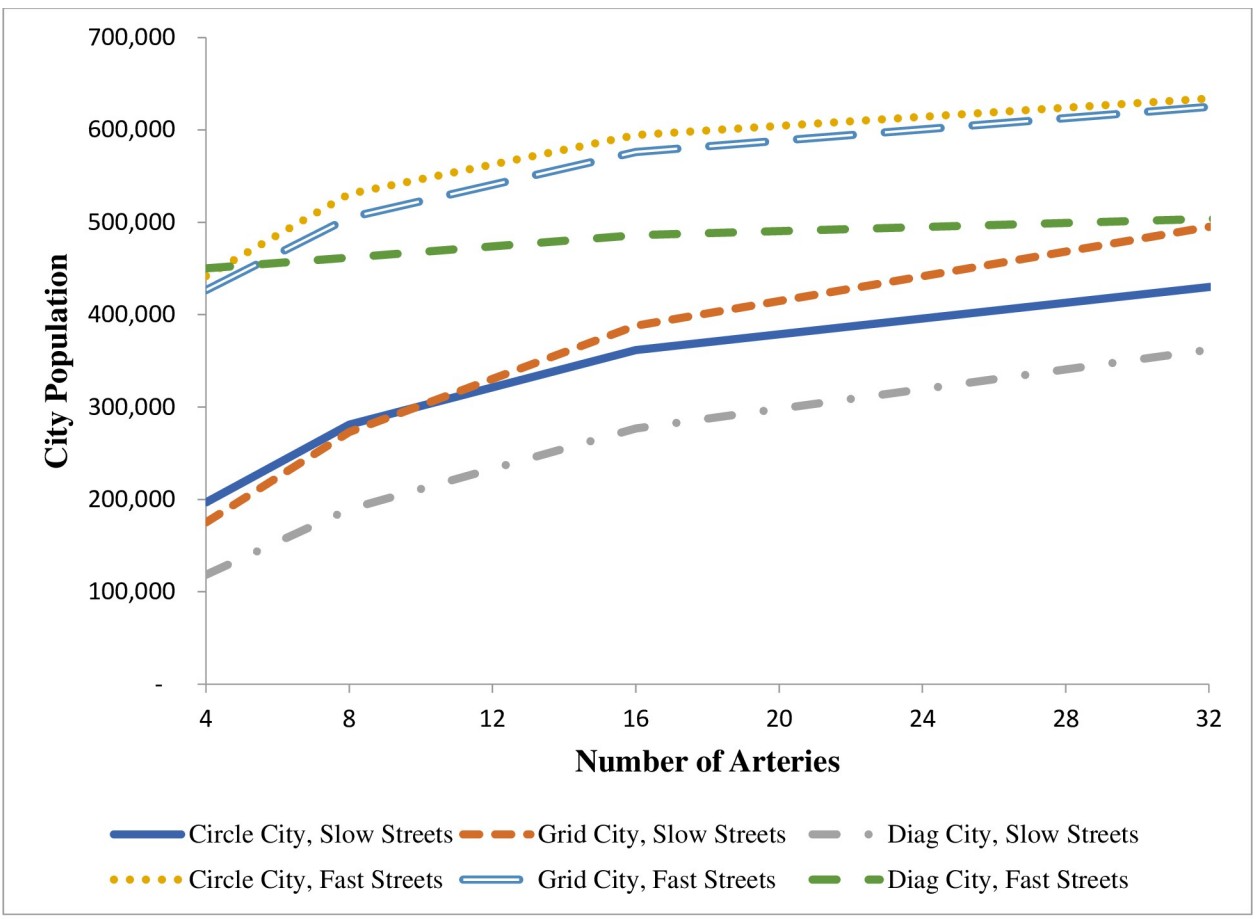

**Fig 7. City population and number of arteries.**

As shown by [8], additional arteries can also be added to a city with a standard grid, but the geometry of this case is complicated. An alternative is to consider cities with grids that are oriented around arteries. This approach leads to an extension of Grid City in which each artery is surrounded by streets that are perpendicular to the artery within the artery's commuting shed and therefore provide access to it. This extension also includes streets that are parallel to the artery but have no role in commuting to the CBD. This arrangement of the street grid continues to the edge of the artery's commuting shed and then rotates to line up with the next artery. The number of arteries in Diag City can be increased by arranging the street grid so that streets intersect each artery at a 45° angle. This design, which has not appeared in the literature, also requires a rotation in the grid at each commuting shed boundary.

Results for the three city types with more than four arteries are provided in Table 1 and in Figs 6 through 8. The associated derivations are presented in the S1 Appendix. The last four columns of Table 1 provide results for cities with sixteen arteries. Each figure shows how one feature of each city changes as the number of arteries changes. Results are given for $\bar{t} = 0.25$ and $\bar{t} = 0.95$. These figures present results based on a doubling of arteries from the previous case. However, the formulas in the S1 Appendix can be applied to any number of evenly spaced arteries. To maintain contact with real cities, these figures stop at thirty-two arteries, which implies a distance of 5.9 miles between arteries at the outer edge (assumed to be 30 miles) of the city.

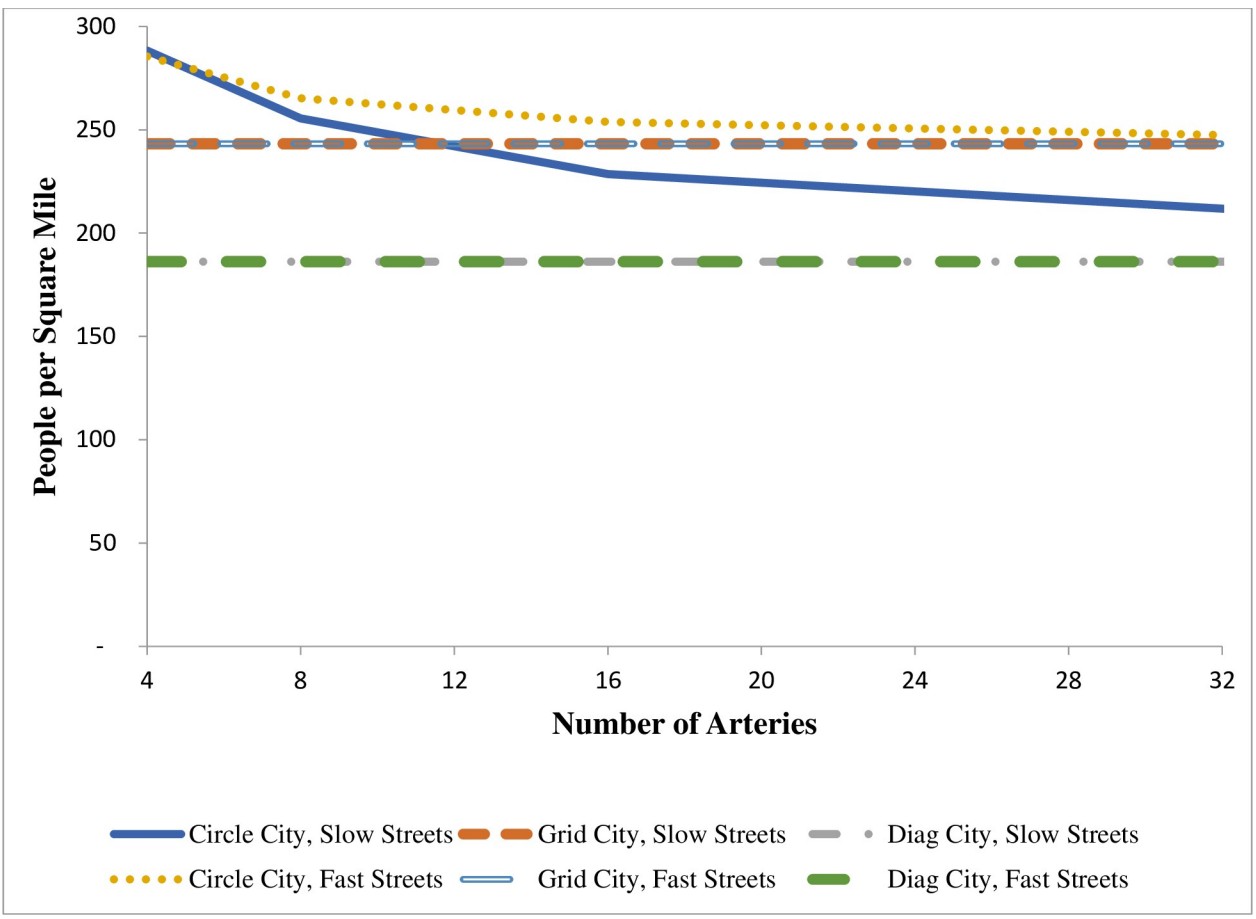

**Fig 8. City population density and number of arteries.**

One key difference between the last four and the first four columns of Table 1 is not surprising: at any given level of $\bar{t}$, adding arteries boosts the area of every city. With a high value for $\bar{t}$, this effect is larger for Circle City and Grid City than for Diag City, and the areas of the three cities converge as the number of arteries grows. See Fig 6. Moreover, Table 1 shows that regardless of the number of arteries, the areas of Grid City and Diag City are the same when $\bar{t}$ = 0.414. In all three cities, the areas slowly converge to the maximum possible area, $\pi(30)^2 =$ 2,827 square miles, as the number of arteries grows. Even with sixteen arteries and $\bar{t} = 0.5$, however, the areas for Circle, Grid, and Diag cities all fall below 75 percent of the area of the basic Ray City with the same value of $\bar{u}$.

Additional arteries also boost the population in all three cities, but at different rates. See the "Population" panel of Table 1 and Fig 7. When $\bar{t}$ is low, additional arteries have a particularly large impact on the population of Grid City, and when $\bar{t}$ is high, additional arteries have little impact on Diag City's population. None of the cities expands far from the arteries when $\bar{t}$ is low, so arteries fill in spaces that fall far toward the CBD and therefore boost population. This effect is particularly strong in Grid City, where, as shown in Fig 4, the iso-cost lines are not "squished" as far toward the CBD as are the iso-cost lines for Circle City. When $\bar{t}$ is high, however, these spaces are much smaller, particularly with the polygon shape of Diag City, so adding arteries does not have as large an impact. In fact, with thirty-two arteries the area of Grid City with small $\bar{t}$ is almost as large as the area of Diag City when $\bar{t}$ is large.

The population density results with many arteries in Table 1 and Fig 8 reinforce the density results with four arteries. Both Grid City and Diag City have constant densities, that is, densities unaffected by $\bar{t}$ or the number of arteries, with a higher density in Grid City than in Diag City. In these two cities the impact of higher $\bar{t}$ or a higher number of arteries on area is exactly offset by their impact on population. (These constant densities are somewhat difficult to see in Fig 8 because the curves for high and low $\bar{t}$ overlap.) The density in Circle city is not constant, however. Instead, density declines as the number of arteries increases, and this decline is greater when the value of $\bar{t}$ is low. In fact, Circle City has the highest density with a high value of $\bar{t}$, regardless of the number of arteries, but its density drops below that of Grid City (but not that of Diag City) when $\bar{t}$ is low and the number of arteries is above twelve.

Increasing the number of arteries not only increases the population of all three cities, it also increases the changes in $\bar{u}$ that are required to bring all three cities up to the same population. When $\bar{t} = 0.25$ and the cities have sixteen arteries, for example, the longest commuting distance in Diag City would have to increase by 1.61 miles (or 5.4 percent) to bring its population up to that of Circle City. See fourth panel of Table 1. Alternatively, one can compare the size requirements needed to reach a given population with each city design. See the last two panels of Table 1. These panels show that the increase in artery length required for a population change from 100,000 to 500,000 is about 12 miles for all city types and for either 4 or 16 arteries. Moreover, adding arteries cuts the value of $\bar{u}$ needed to reach a given population by considerably more when $\bar{t}$ is small than when it is large. This effect is particularly striking for Diag City, which has the smallest impact of arteries on the required $\bar{u}$ when $\bar{t}$ is large and the largest impact when $\bar{t}$ is small.

## 4. Possible extensions

The analysis of cities in this paper could be extended in many ways. This section provides some preliminary thoughts on the use of land for streets, incorporating multiple work sites, and accounting for traffic congestion.

One of the limitations of the continuous mathematics used to solve urban models is that it does not explicitly account for the land area devoted to streets. One partial solution to this problem is to account for the land required to build the commuting arteries. This step is possible because giving a finite width to the arteries does not alter urban model mathematics. In Fig 3, for example, adding width to the arteries does not alter the wedge shape of the residential area between arteries. This wedge-shape preservation applies to all three cities and to any number of arteries. Arteries all meet in the CBD and residential land is a series of wedges between arteries. These wedges can be shifted outward as arteries are added so that the calculations for area and population of the residential parts of the city do not change.

According to [19], the width of a 4-lane urban interstate highway should be at least 100 feet, including 12-foot lanes, breakdown lanes, and medians. Adding more lanes or building a depressed highway with retaining walls and/or access roads obviously increases this width, perhaps to 150 feet or so. Four four-lane artery corridors 100 feet wide and 30 miles long take up 2.27 square miles, which is no more than 0.36 percent of the area of the cities in the first four columns of Table 1. Moreover, sixteen four-lane arteries add up to less than 0.61 percent of the city areas in the second four columns.

It is also possible to make assumptions about the size of a residential lot and the width of streets to estimate the share of the residential area that is devoted to streets. The problem with this approach for this paper is that the street network for Circle City is not comparable to the street network for the other two cities. In fact, non-work trips in Circle City might be quite expensive. A person living on one street must drive all the way to the nearest artery to reach a

destination on another street. If a person lives 15 miles from the CBD in Circle City, she could have a trip to an artery of up to 23.56 miles. With sixteen arteries, this maximum possible trip length drops to 5.95 miles. (The circumference of a circular street 15 miles from the CBD is (2) (15)π = 94.25. Dividing this distance by the number of segments, which is twice the number of arteries, and multiplying by two, because these are maximum round trip costs, yields 5.95.)

The obvious solution to this problem is to build more arteries, which then serve people commuting to the CBD and people making other sorts of trips. This solution is the one in the Black Rock City design. Table 1 and Figs 6 through 8 show, however, that Circle City with many arteries is quite different from one of the grid-based cities with four arteries. In Table 1, Circle City with sixteen arteries has a population density that is similar to the one for Grid City with four arteries, regardless of the value of $\bar{t}$. In these comparisons, however, Circle City is much larger in both area and population. An alternative approach, not considered here, would be to add low-speed arteries to Circle City while retaining a baseline level of high-speed arteries.

A second possible extension is to multiple worksites. Several scholars have introduced an employment beltway into an urban model [20], but to the best of my knowledge, these studies all use the traditional street network, namely, a large number of radial highways. Another approach is to introduce a suburban business district (SBD). [21] assumes radial highways emanating from both the CBD and the SBD and shows that the line separating the residential areas of CBD and SBD workers takes the shape of a parabola. [12] derives the shapes of the residential zones of SBD and CBD workers with the Anas-Moses street network, whereas [8] and [10] derive these shapes with a grid. However, none of these articles compares city outcomes with different street networks.

Finally, future studies of street networks might be extended to bring in traffic congestion. Following in a long tradition (e.g. [22]), [9], [13], and [18] create simulation models that bring congestion into urban models with different street networks. [16] solves an urban model with congestion in the special case of a street grid and vertical arteries to the CBD that are lined up with the grid.

As is well known, the challenge facing this extension is that location choices and congestion are determined simultaneously; that is, people decide where to live based on commuting costs and commuting costs with congestion depend on where people live. Perhaps future research and find a way to model congestion that can be solved with different street networks. Another possibility is to introduce different street networks into an urban model with a bottleneck, such as [23] and [24].

## 5. Lessons for city design

This analysis in this paper leads to several lessons about the design of a city transportation system. These lessons are not just hypothetical. Although a full catalog of street networks in planned cities is beyond the scope of this paper, many cities make use of the designs in this paper.

To begin, Sun City, Arizona, is a series of large neighborhoods with a Circle City design, each with four or five slightly curved arteries [25]. The planned mining town of El Salvdore, Chile, includes half of the Circle City design with many arteries [25]. One large neighborhood in the El Falah Housing Project in Abu Dhabi, takes the form of Circle City with 12 arteries [26].

New York City was not, of course, the first city to use a grid system. Indeed, street grids have appeared in cities since ancient times. One noteworthy example is Philadelphia. A map of downtown Philadelphia from 1683 looks exactly like Grid City with four arteries [27].

Palmalova, Italy, which was built in 1693, looks just like Grid City with 16 arteries [25]. This design was copied by a large neighborhood around Plaza Del Ejecutivo in Mexico City [28]. The semi-circle that makes up downtown Amsterdam is like Grid City with seven arteries [29]. A particularly striking case is Canberra, Australia, which has a Grid City design with multiple arteries on one side of Lake Burley Griffin and a Circle City design with multiple arteries on the other side [25].

Finally, some cities have a grid design with diagonal arteries. As noted earlier, Washington, DC follows this pattern with many arteries, but does not have a rotated grid like the one in Diag City. La Plata, Argentina has a central area designed like Diag City with four arteries [25]. Belo Horizonte, Brazil also uses the Diag City design with four arteries—repeated several times [25].

The first lesson for these city designs reinforces and extends the lesson in previous research that the design of the transportation network influences the shape of a city. Circle City has a curved outer boundary that gives it the appearance of a multi-pointed starfish, with one "point" per artery. Grid City and Diag City are also multi-pointed, but their outer boundaries are straight lines, giving them the appearance of a hand-drawn star. Under some circumstances, Diag City may also be a regular polygon with a vertex for each artery and each commuting shed. With the grid-tilting assumption behind Grid City and Diag City, the shapes of all three cities converge to the circle shape of Ray City as the number of arteries becomes large. However, these shapes remain quite different from each other for any realistic number of arteries.

Additional lessons refer to the three city traits on which this paper focuses: area, population, and average population density. In each case, a city's traits depend on the form of the city's transportation network. Moreover, the traits of the three cities are similar under some circumstances but quite different under others. This general conclusion is not new, of course, but this paper provides the first comparisons using three types of street network and variation in commuting costs.

Consider first a city's physical size. When streets and arteries have similar commuting costs, Diag City is much larger in area than the other two cities. With any other combination of commuting costs and number of arteries, however, all three cities have a similar size. Moreover, as shown by previous studies, cities with relatively fast travel on streets are larger, because households are willing to live farther from an artery the less time it takes to get there.

In the case of population, the analysis in this paper finds that all three cities have similar populations when the number of arteries is small and the value of $\bar{t}$ is high. In all other cases, however, the population of Diag City falls below the population of the other two cities—despite its large area with relatively fast travel on streets. In other words, the population of Diag City does not grow very much as arteries are added because the city already has the high-area polygon form. In addition, adding arteries has a particularly large impact on Grid City's population when arterial travel has a large cost advantage.

Finally, this paper sheds light on the consequences of the street network for average population density. The densities in Grid City and Diag City do not vary with relative transportation costs on streets or the number of arteries, with a higher density in Grid City than in Diag City. Moreover, Diag City has the lowest density of the three cities under all circumstances, and Circle City has the highest density with only four arteries or with relatively fast travel on streets.

Thus, in a city with a market-based land-allocation system, the Black Rock City design (= Circle City) with more than a few arteries has about the same density as the more traditional Grid City, so long as the cost advantages of arterial travel are not large. Planners looking for a high-density city without a large cost advantage for arteries may want to consider both Circle City and Grid City in selecting a transportation network design. Planners who prefer a low-

density city should consider the Diag City design, which best meets this objective, at least among the city designs examined in this paper.

$(0,y^*)$

Black Rock City is built and removed in a few weeks every year. Scholars obviously do not have the opportunity to observe other cities with a similar lifespan but different traits. Nevertheless, scholars can build analytical models to study the impact of street design on urban outcomes. This paper builds urban models for three city types, one of which is based on the Black Rock City street design, and shows, among other things, that this design is ideal for creating a high-density city when arterial travel is not much faster than travel along streets.

## Supporting information

**S1 Appendix. Derivations.**
(DOCX)

## Author Contributions

**Conceptualization:** John Yinger.

**Formal analysis:** John Yinger.

**Investigation:** John Yinger.

**Methodology:** John Yinger.

**Software:** John Yinger.

**Writing – original draft:** John Yinger.

**Writing – review & editing:** John Yinger.

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
