## [Decision Letter · Decision Letter 0]

11 Sep 2020

PONE-D-20-22306

Black Rock City versus Manhattan: An Economist's View

PLOS ONE

Dear Dr. Yinger,

Thank you for submitting your manuscript to PLOS ONE. After careful consideration, we feel that it has merit but does not fully meet PLOS ONE’s publication criteria as it currently stands. Therefore, we invite you to submit a revised version of the manuscript that addresses the points raised during the review process.

We look forward to receiving your revised manuscript.

Kind regards,

Nicolas Chiabaut, Ph.D.

Academic Editor

PLOS ONE

Journal Requirements:

Reviewers' comments:

Reviewer's Responses to Questions

**Comments to the Author**

1. Is the manuscript technically sound, and do the data support the conclusions?

Reviewer #1: Yes

Reviewer #2: Partly

2. Has the statistical analysis been performed appropriately and rigorously? 

Reviewer #1: N/A

Reviewer #2: N/A

3. Have the authors made all data underlying the findings in their manuscript fully available?

Reviewer #1: Yes

Reviewer #2: Yes

4. Is the manuscript presented in an intelligible fashion and written in standard English?

Reviewer #1: Yes

Reviewer #2: Yes

5. Review Comments to the Author

Reviewer #1: The article is well-written. It uses the concrete example of Black Rock City to concretize a new consideration, building upon previous work (e.g., by Anas and Moses). The analysis is sound. Some of it refers to derivations in other works, which is fine. The overall strategy is to demonstrate certain facts using simulation about the difference among aggregates for cities with different street layouts, mainly holding constant either the radius or total population.

Comments:

Section 5 could do with more discussion of the situation in which a planner would be faced with choosing such a design. Black Rock City is a somewhat unusual example in this regard, being rebuilt every year. Mid-century, of course, there were a good number of cities designed and built mostly from scratch, but there are also some in China and India being built today, and the Charter City advocates envision new ones mainly in Latin America and Africa.

It would be illustrative to provide examples of real cities, or parties of real cities, that roughly evoke each class of ideal city.

There is a lot of notation. It would be helpful to have a table of notation, for the reader.

"The objective of this paper is to compare the areas, populations, and average population densities of Circle City, Grid City, and Diag City." I would add something along the lines of "and the maximum distance from the city center to hold a given population." I'm not sure what would be best, but the scope ought to include that the comparisons hold something constant. There are many statements in the paper that could be illuminated by saying what is being held constant as the cities are being compared.

I think there should be some commentary about the generality of the results. We are proceeding mainly via simulation for a few parameter. This isn't to say it's necessary to prove everything.

Reviewer #2: This paper analyzes and compares different city network structures from an economic perspective. While the topic is sound, the paper lacks a clear description of the contributions. The author should better explain the need for each type of the analysis made. In its current form, the paper is difficult to follow. It seems that many parts are build upon the previous studies, thus contributions need to be clearly defined. Since this is an economic analysis, it is useful to incorporate a comprehensive analysis of the transportation cost. There are several studies in the literature comparing different city structures from a public transport perspective that the author might consider incorporating in the paper.

6. PLOS authors have the option to publish the peer review history of their article (what does this mean?). If published, this will include your full peer review and any attached files.

Reviewer #1: No

Reviewer #2: No

---

## [Author Response · Author response to Decision Letter 0]

24 Oct 2020

John Yinger

Black Rock City versus Manhattan: An Economist’s View

Resubmission to Plos One

Responses (in Italics) to Reviewers’ Comments 

Reviewer #1: The article is well-written. It uses the concrete example of Black Rock City to concretize a new consideration, building upon previous work (e.g., by Anas and Moses). The analysis is sound. Some of it refers to derivations in other works, which is fine. The overall strategy is to demonstrate certain facts using simulation about the difference among aggregates for cities with different street layouts, mainly holding constant either the radius or total population.

Comments:

Section 5 could do with more discussion of the situation in which a planner would be faced with choosing such a design. Black Rock City is a somewhat unusual example in this regard, being rebuilt every year. Mid-century, of course, there were a good number of cities designed and built mostly from scratch, but there are also some in China and India being built today, and the Charter City advocates envision new ones mainly in Latin America and Africa.

It would be illustrative to provide examples of real cities, or parties of real cities, that roughly evoke each class of ideal city.

These two comments are very helpful. The first version of the paper contained just a few examples of actual cities with the three analyzed street networks. After an extensive internet search, I was able to identify quite a few more examples of all three street network designs. These examples are provided at the beginning of section 5, “Lessons for City Design.” (See page 17.) In the previous version, a few examples were mentioned in earlier sections. All these examples (except Black Rock City, Manhattan, and Washington, D.C.) have now been moved to page 17. 

There is a lot of notation. It would be helpful to have a table of notation, for the reader.

I have added a table of notation to the appendix. This table could be moved to the text if that seems preferable.

"The objective of this paper is to compare the areas, populations, and average population densities of Circle City, Grid City, and Diag City." I would add something along the lines of "and the maximum distance from the city center to hold a given population." I'm not sure what would be best, but the scope ought to include that the comparisons hold something constant. There are many statements in the paper that could be illuminated by saying what is being held constant as the cities are being compared.

This is a helpful suggestion. I added two panels to Table 1 indicating the city sizes needed to reach selected city populations. A discussion of the results appears on page 14.

I think there should be some commentary about the generality of the results. We are proceeding mainly via simulation for a few parameter. This isn't to say it's necessary to prove everything.

The revised version of the paper adds more emphasis to the point that the simulation results depend on the selected parameters—but the comparisons across cities do not. The two new panels in Table 1 and the revisions to the footnote in Table 1 are directly on this point.

Reviewer #2: This paper analyzes and compares different city network structures from an economic perspective. While the topic is sound, the paper lacks a clear description of the contributions. The author should better explain the need for each type of the analysis made. In its current form, the paper is difficult to follow. It seems that many parts are build upon the previous studies, thus contributions need to be clearly defined. Since this is an economic analysis, it is useful to incorporate a comprehensive analysis of the transportation cost. There are several studies in the literature comparing different city structures from a public transport perspective that the author might consider incorporating in the paper.

I agree with the reviewer that the paper should do a better job highlighting its contributions. I have added a comment on Diag City (p. 12) to explain its contribution. I have retained the statements in “lessons” that explain the paper’s other contributions.

This paragraph from the reviewer also indicates that the paper does not contain a comprehensive analysis of transportation costs. This is a helpful comment, because it points out a key part of the paper that needs clarification. The street/artery framework is quite general. It includes many types of public transportation systems. Page 5 now has a sentence with examples of possible arteries. More complex transportation networks cannot be analyzed with the type of urban model in this paper. I have added a brief comment on a key example of these issues—commuting time—on page 2. I hope these revisions make the paper clearer.

---

## [Decision Letter · Decision Letter 1]

8 Dec 2020

Black Rock City versus Manhattan: An Economist's View

PONE-D-20-22306R1

Dear Dr. Yinger,

We’re pleased to inform you that your manuscript has been judged scientifically suitable for publication and will be formally accepted for publication once it meets all outstanding technical requirements.

Kind regards,

Nicolas Chiabaut, Ph.D.

Academic Editor

PLOS ONE

Additional Editor Comments (optional):

Reviewers' comments:

Reviewer's Responses to Questions

**Comments to the Author**

1. If the authors have adequately addressed your comments raised in a previous round of review and you feel that this manuscript is now acceptable for publication, you may indicate that here to bypass the “Comments to the Author” section, enter your conflict of interest statement in the “Confidential to Editor” section, and submit your "Accept" recommendation.

Reviewer #1: All comments have been addressed

Reviewer #2: All comments have been addressed

2. Is the manuscript technically sound, and do the data support the conclusions?

Reviewer #1: Yes

Reviewer #2: Partly

3. Has the statistical analysis been performed appropriately and rigorously? 

Reviewer #1: N/A

Reviewer #2: N/A

4. Have the authors made all data underlying the findings in their manuscript fully available?

Reviewer #1: Yes

Reviewer #2: Yes

5. Is the manuscript presented in an intelligible fashion and written in standard English?

Reviewer #1: Yes

Reviewer #2: Yes

6. Review Comments to the Author

Reviewer #1: In the response the author raised the idea of moving the table of notation into the paper, rather than the appendix. I think that would be wise.

Another thing I don't think I commented on the first draft: the paper uses curly brackets for function arguments. While this does avoid ambiguity with order-of-operations, I think it is a fairly unusual choice and is confusing. So I would just use parentheses or square brackets.

Other than these two comments I am satisfied.

Reviewer #2: The author has successfully addressed all of my comments. I do not have any further question at this point.

7. PLOS authors have the option to publish the peer review history of their article (what does this mean?). If published, this will include your full peer review and any attached files.

Reviewer #1: **Yes: **Lewis Lehe

Reviewer #2: No

---

## [Editor Report · Acceptance letter]

16 Dec 2020

PONE-D-20-22306R1 

Black Rock City versus Manhattan: An Economist's View 

Dear Dr. Yinger:

I'm pleased to inform you that your manuscript has been deemed suitable for publication in PLOS ONE. Congratulations! Your manuscript is now with our production department. 

Kind regards, 

on behalf of

Pr. Nicolas Chiabaut 

Academic Editor

PLOS ONE